Skin microbiota diversity among genetically unrelated individuals of Indian origin

Potbhare Renuka 1
RaviKumar Ameeta 2
Munukka Eveliina 3 4
Lahti Leo 5
Ashma Richa richaashma@unipune.ac.in 1
1 Department of Zoology, Savitribai Phule Pune University , Pune , Maharashtra , India
2 Institute of Bioinformatics and Biotechnology, Savitribai Phule Pune University , Pune , Maharashtra , India
3 Microbiome Biobank, Institute of Biomedicine, University of Turku , Turku , Finland
4 Biocodex Nordics , Finland
5 Department of Computing, Faculty of Technology, University of Turku , Turku , Finland
Whiteson Katrine
Electronic publication date: 2022 Mar 16
Publication date: 2022
Volume: 10
Electronic Location ID: e13075
Received 2021 Apr 17; Accepted 2022 Feb 16
Copyright: ©2022 Potbhare et al.
Copyright year: 2022
Copyright holder: Potbhare et al.
License: This is an open access article distributed under the terms of the Creative Commons Attribution License, which permits unrestricted use, distribution, reproduction and adaptation in any medium and for any purpose provided that it is properly attributed. For attribution, the original author(s), title, publication source (PeerJ) and either DOI or URL of the article must be cited.
License URL: https://creativecommons.org/licenses/by/4.0/

Keywords: Skin microbiota, Diversity, Geographical variation, Diet, Aging, Indian Population

Funding: Ministry of Human Resource Development (MHRD), Government of India (GOI) Scheme SPARC/2018-2019/P794/SL Academy of Finland 295741 This work was supported by the Scheme for Promotion of Academic and Research Collaboration (SPARC), under the Ministry of Human Resource Development (MHRD), Government of India (GOI) Scheme (SPARC/2018-2019/P794/SL; dated –20/03/2019) and Academy of Finland (decision 295741; to LL). The funders had no role in study design, data collection and analysis, decision to publish, or preparation of the manuscript.

==============================
Background

Human skin harbors complex transient and resident microbial communities that show intra- & inter-individual variation due to various environmental and host-associated factors such as skin site, diet, age, gender, genetics, or the type and use of cosmetics. This variation remains largely uncharacterized in the Indian population; hence, the present study aims to characterize the variation in skin microbiota among individuals of Indian origin and quantify associations with age, diet, and geography.

Methods

Axillary sweat samples from genetically unrelated individuals (N = 58) residing in the three geographical locations of Maharashtra, India, were collected using a sterile cotton swab. Bacterial DNA was extracted using a standard protocol and checked for quality. Variable regions (V3–V4) of the 16S rRNA gene were sequenced using the Illumina platform. We used standard methods from microbiota bioinformatics, including alpha and beta diversity, community typing, and differential abundance, to quantify the association of skin microbiota with age, diet, and geographical location.

Results

Our study indicated the prevalence of phyla- Firmicutes, Proteobacteria, and Actinobacteria, consistent with previous reports on skin microbiota composition of the world population level. The alpha diversity (Shannon index) was significantly associated with the age group (Kruskal–Wallis test, p = 0.02), but not with geography (p = 0.62) or diet (p = 0.74). The overall skin microbiota community composition was significantly associated with geographical location based on Community State Types (CST) analysis and PERMANOVA (R2 = 0.07, p = 0.01). Differential abundance analysis at the genus level indicated a distinctively high abundance of Staphylococcus and Corynebacterium among individuals of the Pune district. Pseudomonas and Anaerococcus were abundant in individuals from Ahmednagar whereas, Paenibacillus, Geobacillus, Virgibacillus, Jeotgalicoccus, Pullulanibacillus, Delsulfosporomusa, Citinovibrio, and Calditerricola were abundant in individuals from Nashik district.

Conclusion

Our work provides one of the first characterizations of skin microbiota variation in different sub-populations in India. The analysis quantifies the level of individuality, as contrasted to the other factors of age, geography, and diet, thus helping to evaluate the applicability of skin microbiota profiles as a potential biomarker to stratify individuals.

Introduction

As one of the largest organs, our skin serves a key role in protecting the body from pathogens, viruses, and toxins present in the external environment (Pasparakis, Haase & Nestle, 2014; Greaves, 2007). Grice et al. (2009) discussed the variation in the taxonomic composition and diversity in microbial communities that occupy the human skin. Human skin microbiota harbors bacteria, fungi, viruses, and mites. The concentration of aerobic bacteria in moist regions like axilla alone accounts for 107 cells/cm2, whereas the density of anaerobic bacteria can count up to 106 cells/cm2 (Fredricks, 2001). The commensal microorganisms that live on the skin prevent colonization and invasion of pathogens and modulate innate and adaptive immunities (Belkaid, 2015).

Human skin harbors resident and transient microbiome species (Nelson, Holder & Maryland, 1938). The resident species can sustain growth, showing relatively stable abundance and composition. These species are present on and within the outermost layer of the epidermis. Such bacterial species remain attached to the skin surface and are long-term body residents. The resident skin microbiota includes genera Cutibacterium (Cutibacterium formerly known as Propionibacterium), Staphylococcus, Micrococcus, Corynebacterium, Malassezia yeast, and bacteriophage species (Scholz & Kilian, 2016; Johnson et al., 2002). The transient species reside primarily on exposed skin and originate from exogenous sources. These bacterial species lie freely on the skin surface and often cannot grow for extended periods in variable physio-chemical conditions, and hence remain on the skin only for a short time. Examples of commonly observed transient species include Escherichia coli, Bacillus species, Staphylococcus aureus, and Pseudomonas aeruginosa (Bojar & Holland, 2002).

Human skin microbiome project based on 16S rRNA profiling from twenty different skin sites representing three skin microenvironment types: sebaceous, moist, and dry, suggested topographical and temporal diversity (Grice et al., 2009). A study by Oh et al. (2016) described that skin’s biogeography in terms of skin site, type, physiology is responsible for the stability and persistence of skin microbial communities. They investigated moist skin regions that were inversely correlated to stability over time. They previously showed that strain-level bacterial composition determines individuality; however, the species-level bacterial composition was more specific to skin physiology (Oh et al., 2014). The most prevalent phyla detected were Actinobacteria (51.8%), Firmicutes (24.4%), Proteobacteria (16.5%), and Bacteroidetes (6.3%) (Grice et al., 2009). Another study on the Chinese population involving 200 skin samples showed the predominance of the same four phyla, with the difference in mean relative abundances in Actinobacteria (36.6%), Proteobacteria (31.6%), Firmicutes (19.1%), and Bacteroidetes (7.1%). These four phyla account for over 94% of detected phylotypes in the Chinese population. The ten most abundant genera, which were well documented and made up over 50% of skin microbiota within each sample, were Cutibacterium, Staphylococcus, Acinetobacter, Streptococcus, Enhydrobacter, and Corynebacterium. Thus, suggesting, geographical and ethnic differences in the human population are the factors that could explain some of the differences observed in the skin microbiota (Leung, Wilkins & Lee, 2015; Ruuskanen et al., 2021).

Fierer et al. (2010) while demonstrating human skin microbial community composition, explained its use in forensics by analyzing residual skin-associated bacteria recovered from touched surfaces. A significant interpersonal variation was observed between left and right antecubital fossae, axillae, and volar forearms. This observation was also supported by Grice et al. (2009), wherein four of the five resampled volunteers showed taxonomic similarity to their previously donated samples (Grice et al., 2009). The significant qualitative and quantitative interpersonal variation on the skin microbiota was due to various driving factors (Roth & James, 1988), such as skin type (Bojar & Holland, 2002), gender (Leung, Wilkins & Lee, 2015; Troccaz et al., 2015; Ross, Doxey & Neufeld, 2017) use of skincare products, perspiring agents (Bouslimani et al., 2019), or age (Capone et al., 2011; Kim et al., 2019). The use of statistical and machine learning techniques to understand the intra- and inter-individual variation and to develop new applications for diagnostic, prognostic, forensic, and related tasks is currently an active research area in human skin microbiome studies (Kyrpides et al., 2021; Moreno-Indias et al., 2021; Zambrano et al., 2021).

Our study aims to complement these previous reports by quantifying the overall taxonomic variation on skin microbiota composition in a previously uncharacterized population of genetically unrelated individuals of Indian origin. We analyzed the association with external factors that may influence skin microbiota composition and inter-individual variability, including age, diet, and geography. We chose three geographical locations that are relatively close to each other, with slight differences in the climate. As all individuals were of Indian origin, we can assess the effect of differences in climatic conditions on skin microbiota taxonomic composition and diversity.

Materials and Methods

Human ethics committee

Human Ethics Committee of Savitribai Phule Pune University, which follows the guidelines of the Indian Council of Medical Research (ICMR), India, approved the present study (Letter no: SPPU/IEC/2019/60 Date: 20/11/2019). After explaining the details of the study, written informed consent was obtained from volunteers.

Subject selection

Before sample collection, the power analysis was carried out using G*Power 3.1.9.4 software to compute the required sample size. The α-level was set at 0.05 (statistical power 95%) with an effect size ρ = 0.50. The total required sample size (N) was calculated to be 57 for this study. Individuals from different housing societies were randomly chosen from three different geographical locations. A summary of the project and study design was provided, and the volunteers filled in the detailed questionnaire concerning health and medication history and lifestyle, including the history of dermatological disease, alcohol consumption, smoking habit, consumption of either antibiotics or long-term medications (diabetes, CVD, hypertension), and extent of physical activity. Volunteers were asked to avoid underarm shaving, bathing, cosmetic items (such as deodorants and perfumes), and certain food items (like onions, garlic, cabbage, and chilies) for 24 h before the sample collection, and this was verified with each person during sampling through observation and communication. Each individual’s dietary habits were recorded in detail (such as frequency of consumption of beef, mutton, fish, and chicken in a week). Only subjects with overall similar dietary habits were recruited for the study. Of the recruited genetically unrelated individuals of Indian origin, there were 11 co-habiting couples.

Sample collection

Sweat samples from genetically unrelated male and female individuals (N = 58, 16 to 91 years), having mixed and vegetarian diets, were collected using a sterile cotton swab. Recruited volunteers belonged to three geographical areas of Maharashtra, India, viz., Pune (altitude: 1,840 ft; longitude:73°51′19.26″E; latitude:18°31′10.45″N), Ahmednagar (altitude:2129 ft; longitude: 74°44′58.53″E; latitude:19°5′42.75″N) and Nashik (altitude:1916 ft; longitude:73°47′27.46″E; latitude:19°59′50.17″N). The demographics of the 58 individuals are represented in Table 1. The three geographical locations are in close vicinity (∼220 kms) to each other, to avoid climate and urbanization differences and their effect on sweat level, we collected the axillary sweat samples within 1 to 2 days. The sterile cotton swab was moistened in sterile PBS solution before sampling, and the volunteers were asked to scrub the moistened swab several times in the axillary region. These cotton swabs (approximately 9 × 7 cm) were placed in a sterile vial to avoid contamination/ degradation, and samples were stored at −80 °C until DNA extraction.

Table 1 The distribution of recruited individuals (N = 58) in different covariates.

The three geographical locations Pune (N = 36), Ahmednagar (N = 11), and Nashik (N = 11). The diet groups: Vegetarian (N = 30) and Mixed (N = 28). Age groups: Adult (16–40 years; N = 24), Middle-age (41–59 years; N = 16), and Elderly (60–91 years; N = 18). Gender: Male (N = 30), Female (N = 28).

Factors	Geographical Location (N = 58)	
		Pune (N = 36)	Ahmednagar (N = 11)	Nashik (N = 11)	
Diet	Vegetarian (%) (N = 30)	61 (N = 22)	64 (N = 7)	9 (N = 1)	
Mixed (%) (N = 28)	39 (N = 14)	36 (N = 4)	91 (N = 10)	
Age
group	Adult (16–40 years)
(%) (N = 24)	25 (N = 9)	46 (N = 5)	91 (N = 10)	
Middle age (41–59 years) (%) (N = 16)	31 (N = 11)	36 (N = 4)	9 (N = 1)	
Elderly (60–91 years)
(%) (N = 18)	44 (N = 16)	18 (N = 2)	0 (N = 0)	
Gender	Male (%) (N = 30)	44 (N = 16)	55 (N = 6)	73 (N = 8)	
Female (%) (N = 28)	56 (N = 20)	45 (N = 5)	27 (N = 3)	

DNA extraction

Bacterial DNA was extracted from sweat samples by Sambrook’s Phenol: chloroform: isoamyl alcohol method (Sambrook, Fritsch & Maniatis, 1989). Extracted bacterial DNA samples were measured on the Nanodrop spectrophotometer at 260 nm and 280 nm absorbance. Further, these DNA samples were diluted to 25 ng/ul, of which 5 ng/ul was used for 16S rRNA sequencing of V3 and V4 region on Illumina miSeq V2 standard flow cell technology.

16S rRNA gene amplicon sequencing

Amplification of the V3–V4 region of 16S rRNA with specific forward and reverse primers was carried out using KAPA HiFi HotStart Ready Mix® (KAPA biosystems, Boston, MA, USA), following manufacturer’s amplification mix composition and thermal cycling conditions (Klindworth et al., 2013). PCR products were purified using PureLink™ PCR purification kit® (Invitrogen, Waltham, MA, USA). The purified PCR products were then ligated with a specific barcode adapter using the Nextera XT Index Kit® (Illumina, San Diego, CA, USA) following the manufacturer’s amplification mix composition (Illumina, 2013). Indexing PCR products were purified using PureLink™ PCR purification kit® (Invitrogen, Waltham, MA, USA). The amplicon concentration was assessed using Qubit 2.0 fluorometer (Life Technologies, Waltham, MA, USA) using Qubit™ dsDNA HS Assay Kit® (Invitrogen, Waltham, MA, USA) following manufacturer’s protocol, and samples were stored at −20° C until further use. A MilliQ water sample was used as negative control and a healthy gut sample of a known microbial community as a positive control. Controls were amplified in the same run to monitor contamination and next-generation sequencing (NGS) run.

Illumina library preparation and sequencing

The quantified samples were normalized by diluting them to 4 nM. An aliquot of diluted DNA (5ul) was loaded on Miseq V2 standard flow cell, and quantitation of DNA samples was redone for validating DNA concentration. qPCR was performed using the PerfeCTa® NGS library quantification kit for Illumina® sequencing platforms (Quanta Biosciences, Maryland, USA). High-throughput sequencing on a MiSeq instrument (Illumina, San Diego, CA, USA) was used to analyze the amplified V3–V4 region of the 16S rRNA. According to the manufacturer’s protocol, the MiSeq reagent v3 kit (500 cycles) (Illumina, San Diego, CA, USA) was used to prepare the final quantified pool for sequencing. A 5% PhiX control was included to increase sample diversity (Illumina, San Diego, CA, USA).

Assembly of sequence data

Microbiome sequence processing was carried out in three steps: preprocessing, clustering similar sequences to obtain Operational Taxonomic Units (OTUs), and annotating the OTU sequences (Sandhu, Pourang & Sivamani, 2019). The raw FASTQ files were filtered and trimmed at Q score 30 to remove low-quality reads and PCR-generated chimeric sequences. We used the method from (Illumina, 2014), to generate the OTU table based on taxonomic identification with the Greengenes database (DeSantis et al., 2006). A total of 955 206 reads (ranging from 836 to 101 182 reads) of >250 base pairs passed a quality score (Q) of 30. These were clustered into 2,569 OTUs based on a <98% sequence identity cut-off. The OTUs were further refined using data normalization, wherein we removed singleton OTUs.

Statistical analysis

Analysis of skin microbiota composition was carried out with the phyloseq (McMurdie & Holmes, 2013) and microbiome (Shetty & Lahti, 2019) R/Bioconductor packages (R Version 3.6.1). We compared the alpha and beta diversities between volunteer groups based on age, diet, and geography. The Shannon diversity index was chosen for alpha diversity analysis. Group-wise alpha diversity comparisons were performed at the genus level with the Kruskal–Wallis test (Wilcoxon for two-group comparisons). Beta diversity was quantified with Bray–Curtis dissimilarity, and differences in community composition between predefined groups were quantified with PERMANOVA (99 permutations) (Anderson, 2017) using the R vegan::adonis function (Oksanen et al., 2020). Principal coordinates analysis (PCoA) based on the Bray-Curtis dissimilarity index was used to visualize community-level similarity. Community state types (CSTs) were identified based on partition around medoids (PAM) clustering using the cluster R package (Maechler et al., 2021). Clusters were chosen based on visual investigation of the clustering scores as in DiGiulio et al. (2015). The association between the CSTs and the discrete groupings (age, diet, geography) was quantified using Fisher exact test. To further quantify the association of abundant genera within the geographical location, an analysis of compositions of microbiomes (ANCOM) was performed (Mandal et al., 2015). The Dunn post hoc test was performed using the Kruskal–Wallis test for pairwise multiple comparisons on subgroups to compare median similarities of the genus within individuals. The p-values were adjusted and reported using the Benjamini-Hochberg (BH) method.

Results

Core microbiota

We observed altogether 26 bacterial phyla, 53 classes, 119 orders, 234 families, and 673 genera in whole study population. The four most prevalent phyla had >20% prevalence above the 0.1% abundance threshold Firmicutes (mean relative abundances 59.8%, prevalence 100%), Proteobacteria (24.4%, 94.8%), Actinobacteria (15.4%, 96.6%) (Table 2). Four most prevalent genera were Staphylococcus (mean relative abundances 23.2%, prevalence 100%), followed by Bacillus (21.7%, 100%), Corynebacterium (7.1%, 93.1%), and Anaerococcus (5.8%, 77.6%). Along with these genera, we observed 20 genera exceeding 0.1% relative abundance and >20% prevalence threshold (Figs. 1A–1B, Table 3). We consider the remaining 649 unique genera as accessory microbiota since they were absent in most individuals. Further, the relative abundance for the individuals in each geographical location at the phylum level is shown in Fig. 1C.

Table 2 The most prevalent phyla observed with mean relative abundance, and standard deviation.

Prevalence (above the detection threshold = 0.1%). The phyla with >20% prevalence are shown.

Phyla	Mean relative abundance (%)	Standard deviation (σ)	Prevalence (%)	
Firmicutes	59.8	0.37	100.0	
Proteobacteria	24.4	0.33	94.8	
Actinobacteria	15.4	0.24	96.6	

Figure 1 Relative abundance of the most prevalent phyla and genera observed on the skin microbiota of the genetically unrelated individuals (N = 58) of Indian origin.

(A) The relative abundance and (B) prevalence for the most prevalent genera observed on the skin microbiota of the genetically unrelated individuals (N = 58) of Indian origin. The genus names are sorted by prevalence in both figures. The mean relative abundance and prevalence values are shown in Table 3. (C) Phylum level relative abundances in each volunteer that vary most between the three geographical locations (i) Ahmednagar (ii) Pune (iii) Nashik.

Alpha diversity

Host and environmental factors influence skin microbiota composition. We compared skin microbiota alpha diversity (Shannon index) with geographical location, age, and diet. Statistically significant differences were not observed in alpha diversity between neighter the geographical locations (Kruskal–Wallis test, p = 0.62, Fig. 2A) nor the diet groups (Wilcoxon test, p = 0.74, Fig. 2B). In contrast, we did observe a significant difference between the three age groups (Kruskal–Wallis test, p = 0.02, Fig. 2C). Further, multiple pairwise comparisons between each two age groups were performed using the Wilcoxon test. Statistically significant differences were noted between the adult and middle age (p = 0.001) groups. We did not observe differences in alpha diversity between the elderly age group and adults (p = 0.715) and or middle-aged (p = 0.125) volunteers. The group-wise average alpha diversities for geographical location, diet, and age are shown in Table 4.

Table 3 The most prevalent genera along with its mean relative abundance and standard deviation across genetically unrelated individuals (N = 58).

For abundance, detection threshold = 0.1% was considered. The genera with >20% prevalence are shown.

Genus	Mean relative abundance (%)	Standard deviation (σ)	Prevalence (%)	
Staphylococcus	23.2	0.34	100	
Paenibacillus	1.3	0.07	100	
Bacillus	21.7	0.33	100	
Corynebacterium	7.1	0.15	93.1	
Pseudomonas	2.3	0.07	86.2	
Thiolamprovum	0.1	0.00	86.2	
Clostridium	0.2	0.01	86.2	
Arthrobacter	1.3	0.08	79.3	
Anaerococcus	5.8	0.16	77.6	
Oceanobacillus	1	0.04	77.6	
Cutibacterium	1.6	0.08	74.1	
Acinetobacter	2.4	0.10	72.4	
Brevibacillus	0.7	0.03	72.4	
Intestinimonas	0.2	0.01	67.2	
Aeromonas	0.3	0.01	65.5	
Salmonella	1.7	0.07	63.8	
Moraxella	2	0.05	62.1	
Geobacillus	0.7	0.02	62.1	
Brevibacterium	0.2	0.01	60.3	
Pantoea	2.5	0.13	56.9	
Enterobacter	6.6	0.18	56.9	
Virgibacillus	0.4	0.01	56.9	
Micrococcus	0.8	0.04	55.2	
Exiguobacterium	1.2	0.07	46.6	

Figure 2 Alpha diversity variation observed on the skin microbiota across studied groups.

Alpha diversity variation (Shannon index) for the N = 58 study volunteers by (A) geographical location (p = 0.62; Kruskal–Wallis), (B) diet (p = 0.74; Wilcoxon test) and, (C) age group (p = 0.02; Kruskal–Wallis). The p-value indicates level of significance as <0.001∗∗∗, <0.01∗∗, <0.05∗, 0.1ns. Statistical differences are indicated only for the significant groups (Table 4).

Table 4 The alpha diversity for each group of covariates.

The mean Shannon diversity and standard deviation for each group of covariates are shown (N = 58).

Factors	Mean Shannon diversity
(x ¯)	Standard deviation
(σ)	
Geographical location	Pune	1.18	0.69	
Ahmednagar	1.48	0.88	
Nashik	1.27	0.73	
Diet	Vegetarian	1.24	0.65	
Mixed	1.27	0.82	
Age group	Adult (16-40 years)	1.45	0.66	
Middle age (41–59 years)	0.89	0.49	
Elderly (60–91 years)	1.31	0.90	

Beta diversity

Further, we quantified the associations between skin microbiota composition and geography, age, and diet (see Methods). Significant differences were observed in community composition across the three geographical locations (p = 0.01), but no significant differences were observed with age (p = 0.24) or diet (p = 0.55) parameters (Table 5). The PCoA ordination (Figs. 3A–3C) and comparison with the detected community types indicated that the clusters are partially associated with the geographical locations (Fig. 3D). Furthermore, the individuals were categorized into three broad community state types (CSTs) (DiGiulio et al., 2015). CST is a standard cluster analysis for microbial community analysis wherein each CST represents one community type, with a peculiar community composition shared by individuals who fall into that cluster. We observed a significant association between the CST clusters and geography. We have numbered the CSTs from one to three; each CST is abundant in a different set of taxonomic groups. We have visualized the community composition for each CST on a heatmap (Fig. 4). The genera that differ most between the CSTs were Tuberibacillus, Treponema, Staphylococcus, Pullulanibacillus, Halobacillus, Geobacillus, Desulfosporomusa, Bacillus, Anoxybacillu, and Anaerococcus.

Table 5 Association of skin microbiota composition (beta diversity) with key covariates (PERMANOVA; Bray–Curtis index).

The p-values are based on 99 permutations.

Factors	Df	R2	p-value	
Geographical location	2	0.07	0.01**	
Age group	2	0.03	0.24	
Diet	1	0.01	0.55	
Notes.

*** <0.001

** <0.01

* <0.05

ns <0.1

Figure 3 Observed beta diversity on skin microbiota composition using Principal Coordinates Analysis.

Taxonomic similarity (beta diversity) of skin microbiota composition among the study volunteers illustrated on Principal Coordinates Analysis (PCoA; Bray–Curtis). The samples are colored according to; (A) diet; (B) age group; (C) geographical location (D) community state type (CST).

Figure 4 Taxonomic composition by community state type.

Taxonomic composition by community type (CST)(A–C). CST is a clustering method for analyzing bacterial community compositions wherein, individuals with a similar taxonomic composition cluster together. Here, columns correspond to individuals and the rows correspond to genera. The colours indicate the clr/z-transformed abundance of each bacteria compared to its own population mean of the taxa. The abundances have been scaled to zero mean and unit variance to allow visual comparison between taxonomic groups with different abundance levels.

Differential genus abundance

Our analysis revealed twelve genera being significantly associated with the three geographical locations. Pairwise comparisons using the Dunn test indicated a high abundance of Staphylococcus and Corynebacterium on the skin of individuals from the Pune district (Figs. 5A and 5C). A high abundance of Paenibacillus, Geobacillus, Virgibacillus, Jeotgalicoccus, Pullulanibacillus, Delsulfosporomusa, Citinovibrio, and Calditerricola, was observed on the skin of individuals from the Nashik district (Figs. 5B and 5F–5L). In Ahmednagar individuals, Pseudomonas and Anaerococcus were observed in abundance (Figs. 5D and 5E). These variations in the taxonomic compositions across three locations could be due to differences in urbanization and climate, the effect of which was otherwise minimized during sampling.

Figure 5 Genus abundances in geographical locations arranged as per most abundant in a panel.

Pune (A, C), Nashik (B, F–L), and Ahmednagar (D–E). The black line represents the median and the boxes represent interquartile ranges. The p-values are based on the Dunn test using the Kruskal-Wallis test and adjusted with the “Benjamini–Hochberg (BH)” method. The p-value indicates level of significance as <0.001∗∗∗, <0.01∗∗, <0.05∗, <1ns.

Discussion

Research on skin microbiota is an emerging field, which would help to understand the significance of skin microbiota in health, disease, and its inter-and intra-individual variations. High-throughput sequencing based on 16S rRNA gene has generated a broad range of bacterial community analyses. In 2016, Meisel et al. suggested using the V1–V3 hypervariable region of 16S rRNA for skin microbiome studies, as these were the most accurate recapitulation technique for Cutibacterium. However, a recent study by Castelino et al. (2017), after comparing the results of the V1–V3 and V3–V4 amplicons on two sequencing platforms MiSeq and 454, found V3–V4 providing better capture of healthy skin microbial diversity irrespective of the sequencing platform.

Healthy individuals harbor a typical skin microbiota composed mainly of phyla Firmicutes, Proteobacteria, and Actinobacteria. The Review of Byrd, Belkaid & Segre (2018) discussed the previous studies on site-specific skin microbial composition of healthy individuals and reported the predominance of Corynebacterium and Staphylococcus genera on the moist body regions similar to our findings (Byrd, Belkaid & Segre, 2018). Same genera were also reported in the skin samples of healthy individuals from the USA as part of the human microbiome project (Huttenhower et al., 2012). Further, Li et al. (2019) have also revealed the predominance of Staphylococcus and Corynebacterium in healthy subjects. They found dominance of phyla Firmicutes, Proteobacteria, and Actinobacteria in addition to Bacteroidetes and Fusobacteria (Li et al., 2019). Taylor et al. (2003) showed the dominance of Corynebacterium and Staphylococcus with lower levels of Cutibacteria in the axilla of healthy individuals of the U.K. Another axillary skin microbiota study on 53 healthy individuals reported an abundance of Actinobacteria, Firmicutes, and Proteobacteria with the predominance of Corynebacterium and Staphylococcus genera (Callewaert et al., 2013). However, in contrast to earlier studies showing the dominance of phylum Actinobacteria (Schommer & Gallo, 2013), we observed Firmicutes and Proteobacteria in the Indian individuals. Our observations are aligned with another recent study on the skin microbiota of 54 healthy individuals from six Indian families, which showed the prevalence of Firmicutes, Proteobacteria, and Actinobacteria (Chaudhari et al., 2020). Similarly, a study on facial microbiota of healthy females (N=30) from Bangalore (India) detected dominance of four phyla viz., Actinobacteria, Firmicutes, Proteobacteria, and Bacteroidetes (Mukherjee et al., 2016). Earlier, Grice et al. (2009), found an abundance of Staphylococcus and Corynebacterium in the axillary region. The study also emphasized that skin microbiota diversity and variation depend on body site (sebaceous, moist, and dry). A comparative skin-microbial study on primates and human axilla has also been done, which suggested that humans and monkeys share 95% of core microbiome taxa. However, a relatively large microbial diversity was detected in monkeys resembling their soil habitat, whereas the dominance of Staphylococcaceae was found in human axilla (Council et al., 2016).

The transient environment of skin is influenced by the contact with other surfaces and surroundings (Fierer et al., 2010). The climatic condition of a geographical area is determined by humidity, temperature, rainfall, and sun heat which influences the sweat level, affecting skin microbiota composition and diversity. The previous studies by Ross et al. (2018) have shown that the geographic location affects the diversity of mammalian skin microbiota. They further interpreted that accurate classification of mammals could base on their geographical location, which might be different because of the type of soil present in a given habitat. Hospodsky et al. (2014) compared left and right-hand microbiota of women from the US and Tanzania and showed the prevalence of Propionibacteriaceae, Staphylococcaceae, and Streptococceacae in the former and the predominance of Rhodobacteraceae and Nocardioidaceae in letter. These earlier studies re-enforce our finding of variation in the skin microbiota due to geographical location. The three geographical locations selected in the present study are not very far off, but there is a slight difference in the climatic condition. We wanted to see if limited variation in climatic condition affects distribution, composition, and abundance of skin microbiota within a group of individuals with a relatively homogeneous genetic and cultural background. Our analysis suggested statistically significant differences in skin microbiota composition in the three geographical locations. Although geographical location-wise sample distribution is uneven/low, we found robust results and clear association using widely used microbiome statistical analysis methods. These variations, therefore, could be due to varying climatic conditions in the selected three cities. Of the three locations, Nashik has the hottest and driest climate; in Ahmednagar, the climate is hotter and drier than Pune.

Similarly, the average humidity and rainfall received in Nashik (67%, ∼99.3 mm) are higher compared to those in Pune (60.3%, ∼67.9 mm) and Ahmednagar (59%, ∼46.3 mm). A recent study by McCall et al. (2020) has suggested alterations in skin microbiome diversity due to urbanization and lifestyle-associated changes in an individual. Our results indicating differences in skin microbiota composition among the individuals of three geographical locations could also be due to urbanization status and population size per city. Pune has the highest population size (∼.3.1 million), followed by Nashik (∼1.6 million) and Ahmednagar (∼0.3 million). Likewise, Pune, a metropolitan city, has developed industrial areas and infrastructure, and individuals residing in Pune are adapted to urbanization and the western lifestyle. They use skin ointment and cosmetics like moisturizers, deodorants, and antiperspirants, altering axillary bacterial communities. The axillary studies of Grice et al. (2009), Kim et al., 2019, Callewaert et al. (2013), and Council et al. (2016) found dominance of Corynebacterium and Staphylococcus genera in individuals residing in cities which complies with our results of differential abundance analysis. Our analysis indicated that both Corynebacterium and Staphylococcus genera were higher in Pune than the other two studies areas, although we ensured that volunteers did not apply deodorants and cosmetics to their skin/axilla 24 h before sample collection. In contrast, a moderate level of urbanization is found in the Nashik and relatively low in Ahmednagar, where the poor establishment of industrial units and development of agricultural land is observed. Ahmednagar individuals hence are more exposed to soil or environment than populations in other two cities. Urban individuals with more indoor activity predominantly showed a human-derived community (Ying et al., 2015). Earlier studies comparing volar forearms and back of the hand bacterial composition of individuals from urban and rural areas of China showed an abundance of Trabulsiella in the urban population compared to the rural population. These differences could be due to direct contact with soil bacteria or other environmental factors, as studied adults and elderly individuals of the rural area were active workers in the agricultural field. Despite the limited sample size from geographical locations, we could identify a robust and significant association with taxonomic composition.

Skin is a regenerating organ that sheds off the cells, regrows, and renews itself. Microbial colonization of skin starts at birth and continues with age and is affected by habitat, environment, and lifestyle. A study by Bokulich et al. (2016) suggested that the birth mode (cesarean or vaginal) shapes skin microbiota. They found that babies’ early skin microbiota composition (1 and 10 years) is more similar to their mother’s than unrelated women’s (Bokulich et al., 2016). In order to investigate the effect of age on skin microbiota composition, we categorized the volunteers into three age groups. A significant association was observed between the alpha diversity and age, particularly between young and middle-aged individuals. A significant difference in beta diversity was not observed. Present observations agree with the study by Chaudhari et al. in 2020, where no significant difference was observed in alpha diversity of skin microbiome-based on three age groups (Chaudhari et al., 2020). Somerville reported the age-dependent distribution of microbial genera in 1969. He explained that infants have lower diversity because of less environmental exposure than their mother; infants tend to have microbiota similar to their mother (Somerville, 1969). Children coming in contact with people and exposed to the outside environment have more diverse microbiota than infants. At puberty, as the sebum production starts, it further affects the composition of the microbiota. At this stage, diversity reaches its peak; however, it remains stable or decreases with age (Leyden et al., 1975). Kim et al. (2019), determined and compared age-wise alpha diversity and showed significantly high diversity in adults (20–30 years) than in elderly individuals (50–60 years). Furthermore, species evenness was found to be lower in the younger group. They also found that Bacteriodetes (Prevotella, Prophyromonas, Spingobacterium) were significantly abundant in the younger age group, whereas Proteobacteria and Actinobacteria (Micrococcus, Corynebacterium, Dermacoccus) were more abundant in the elderly group.

Nutrition level is known to affect the biochemical and physiological parameters of the human body and, subsequently, skin microbiota composition. Several clinical and therapeutic implications had been characterized by studying gut-skin microbiome and metabolome (Bouslimani et al., 2015). Our study did not find significant differences between the two dietary groups regarding microbiota diversity or taxonomic composition. However, the more subtle effects might have been overdriven by the dominant role of geographical variation with the limited sample size of 58 individuals in our study. Higher diversity of skin microbiota in rural populations has been reported earlier, for instance, by Clemente et al. in 2015, who compared skin microbiota between American-Indians and Western individuals (Clemente et al., 2015). A study by Sriharsha et al. (2015) suggested that diet can contribute to skin health. Nutrients are metabolized and excreted through the skin as sweat metabolites are converted to odoriferous compounds by skin microbial communities. Therefore, diet alterations can eventually lead to skin dysbiosis. Their studies have shown that the diet is associated with skin disease conditions like atopic eczema, dermatitis herpetiformis, acne, and psoriasis. Although several studies show the association of gut microbiome with diet, a correlation between skin microbiome diversity and diet is less studied. Recently, probiotic-based skin ointments for atopic dermatitis therapeutics have yielded positive results implying a relationship between diet-dependent gut-skin microbiome axis (Sherwani et al., 2018).

While our study is informative for subsequent study design and extensions, a larger sample size would be helpful for additional verification and analysis. The co-habitation of the individuals and differences in the sample sizes between age groups can potentially add confounding effects and reduce the statistical power of the analysis, respectively. Profiling environment bacterial communities of the three geographical locations is another potential limitation of the present study, which could have provided valuable and better insight on differences in the skin microbiota of genetically unrelated individuals. Our study helps to establish the basis for designing future studies.

Geography, age, diet, gender, and ethnicity have previously contributed to skin microbiota composition. Our findings have highlighted the notable contribution of geography, intimately associated with various environmental and cultural differences, as a significant driver of the observed variation in community composition.

Conclusion

The human skin harbors a diverse and variable microbiota with remarkable inter-individual differences due to host and environment-associated factors. We observed significant community composition differences at the genus level between the three geographical locations, implying it is as a significant driving factor for community composition and variation. This is the first study based on skin microbiota profiling of genetically unrelated individuals of the same ethnicity from three different geographical locations in the Indian population to the best of our knowledge. Outcomes of the present study can partially inform future research on the Indian skin microbiota. This study’s extension could characterize individuality and variations across the Indian population to better understand the interplay between cultural, environmental, and genetic factors.

Supplemental Information

Supplemental Information 1 The detected total number of bacterial reads (abundances) as per Operational Taxonomic Unit (OTUs) across each sample

Click here for additional data file.

Supplemental Information 2 Bacterial taxonomic classification

List of bacterial taxonomic classification based on obtained sequences with respect to Operational Taxonomic Unit (OTUs).

Click here for additional data file.

The authors thank the Department of Zoology, Savitribai Phule Pune University, for providing infrastructure and equipment facilities. Authors are grateful to the volunteers for their participation in this ongoing study. Authors thank Mr. Manoj Vaikar for sample collection and DNA isolation.

Additional Information and Declarations

Competing Interests

Author Contributions

Human Ethics

DNA Deposition

Data Availability

The authors declare there are no competing interests.

Renuka Potbhare performed the experiments, analyzed the data, prepared figures and/or tables, authored or reviewed drafts of the paper and approved the final draft.

Ameeta RaviKumar, Eveliina Munukka and Richa Ashma conceived and designed the experiments, authored or reviewed drafts of the paper, and approved the final draft.

Leo Lahti conceived and designed the experiments, analyzed the data, prepared figures and/or tables, authored or reviewed drafts of the paper, and approved the final draft.

The following information was supplied relating to ethical approvals (i.e., approving body and any reference numbers):

The present study has been approved by the Human Ethics Committee of Savitribai Phule Pune University (Letter no: SPPU/IEC/2019/60 Date: 20/11/2019), which follows the rules of the Indian Council of Medical Research (ICMR), India.

The following information was supplied regarding the deposition of DNA sequences:

The samples used here are available at the European Nucleotide Archive: ERS6234946 to ERS6235003; PRJEB44216.

The following information was supplied regarding data availability:

The raw data are available in the Supplementary Files.

The source code for the analysis is available under the open-source MIT license: Renuka Potbhare, & Leo Lahti. (2021). Article_2021_sparc (1.01). Zenodo. https://doi.org/10.5281/zenodo.5805004.

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
