# Peer review of "Skin microbiota diversity among genetically unrelated individuals of Indian origin"

_PeerJ, doi:10.7717/peerj.13075_

## Round 0.1 · original submission · Major Revisions

Thank you for your interesting study and manuscript. Both reviewers have provided constructive criticism to improve the visualization and interpretation of the data. With careful attention to these comments, improved figures and re-writing, the manuscript will be acceptable for publication. It is important not the overstate what can be learned from these data - for example, the V3-4 region of 16S is known to miss major skin taxa, please make sure these limitations and caveats are clearly stated in the manuscript.

Reviewer 1 ·

Basic reporting

English language is good.
Some important literature is missing in the manuscript
The figures could have been better, fi. the bacterial composition in each region.

Experimental design

Number of samples is rather low.
more info - see below

Validity of the findings

see below

Additional comments

Authors described the underarm bacterial microbiome of Indian people in 3 distinct locations of India. Using amplicon sequencing of the V3-4 region of 16S rRNA. The author found clear associations with location. No impact of age or diet were found.

Comments

Minor comments
There have been other papers describing the underarm microbiome (eg. Callewaert et al., 2013). It would be meaningful to compare your results with previous results.

Would there be overlap between environmental bacteria and the skin microbiota? Is this a driving factor in the differences seen per location? Can authors hypothesize on this?

What is the Impact of gender? It is not reported in the manuscript, although in the discussion section, it was mentioned as not significantly associated?
Can authors include gender in Table 1 (or another table)?

Can the authors describe the characteristics of Pune, Ahmednagar, and Nashik? Are these big cities? How many inhabitants? I had to go look it up online: Pune (3000000), Ahmednagar (350000), and Nashik (1500000). How is rainfall, temperature, humidity etc different in each city? It is generally discussed in the discussion, but reader has no clue on the differences for each city.
Does skin diversity associate with city size (inhabitant number)?

Can the authors describe more about urbanization grade in each location? How are Pune, Ahmednagar, and Nashik different from each other? Urbanization paper: McCall et al, 2020

Age: no correlation found; but do the authors see pattern with sebum / apocrine sweat production (which is the highest at ~16-30y)? Are there more lipophilic bacteria around that age?

N=58 might be low. Did the authors perform a power calculation to verify if the number of samples is sufficient for each location?


Major comments
The V3-V4 region of the 16S rRNA region was amplified. It is known that this region gives an underrepresentation of the Propionibacterium/Cutibacterium genus (Meisel et al., 2016 JID); which are relatively important taxa on the human skin and in the underarms.

Cutibacterium is not mentioned once. Propionibacterium has been renamed to Cutibacterium recently (Scholz et al., 2016 Int. J. Syst. Evol. Microbiol).

Figure 1 should be a bargraph of the underarm microbiome of the 3 different locations. The current log scale makes it really difficult to interprete. Also, what are the other taxa?? This is not the best way to represent the bacterial composition.

Figure 2 is of poor quality. Can the author add/visualize statistics to the plots?

Figure 4 is again of poor quality.
How have the 10 taxa been chosen; is this the top 10 most abundant taxa, and then ranked according alphabet? I guess not, because Corynebacterium is missing.
Suggest to rank according abundance: so Staph first. Now it seems as if Tuberibacillus is the most important genus in the underarms of Indian people. Which I highly doubt, and to be honest, I cannot even know what are the most abundant taxa; because it is not reported ) (except the top 4 most abundant in Fig 1).

I would argue that Nashik, Ahmednagar and Pune are not so distinct from each other. They are actually geographically quite close to each other. In the same province / mid-West of India. What is the reasoning behind the choice for these 3 areas?
I suggest authors describe in much more detail why these 3 cities are so different from each other.



All in all an interesting paper, but it’s really hard to comprehend and interpret the results. No clear figure on bacterial composition; unclear why the 3 locations were chosen, which gender the volunteers were, etc.

Reviewer 2 ·

Basic reporting

Line 62: "The commensal....pathogens...immunities" requires citation

Lines 202-203 Indicate that analysis is available on github; however the provided link is not accessible.

Experimental design

Impact of diet on skin microbiome is less explored albeit an important area of exploration. The diets should be defined in more depth to make meaningful interpretation. Additionally, since the authors do review the literature in significant depth, it would be useful to discuss findings from other relevant studies as well.

In sample collection, authors’ state:
“To avoid the effect of geographical location and related climatic differences, the axillary sweat samples were collected within 1 to 2 days from all the volunteers.” It is unclear as a reader, what does “within 1-2 days of what reference point?” mean.

While the representation of figures in the microbiome field is always up to debate, it is always useful to have pictorial/graphical depictions of data wherever possible. It would be useful to have some of the tables, particularly Tables 3 and 4 as graphs. Further, some of the figures are of lower resolution, making them hard to read.

Validity of the findings

Please refer to general comments for detailed points.

Additional comments

The manuscript by Potbhare, et al (2021) is an important manuscript in the context of the Indian subcontinent as there are not many studies that investigate the composition of the skin microbiome in the defined population. The study samples 58 volunteers, of which 36 belong to Pune, making the data from Ahmednagar (n=11) and Nashik (n=11), slightly over-interpretative due to small sample sizes. Generally, the manuscript is written clearly, and the literature reviewed is appropriate in the context of skin microbiome. Since the first description of skin microbiome using culture-independent technique, several advances have been made in the field. The authors use 16s rRNA profiling to sequence the samples, which is fairly adequate in the context of this manuscript. However since then studies performing whole-metagenomic sequencing [Oh, et al., 2016] have described the microbiome at a deeper resolution. It is important to discuss these findings as well. The other issue, I have as a reviewer is that the authors’ only sample the axillary sweat region. The axillary sweat region consists of hair follicles, sebaceous glands and is rich in sweat glands as such making it a completely unique microenvironment that is not exactly representative of the skin microbiome. While the authors’ discuss the differences in body site, this important point is not discussed in great depth. As of current, the axillary microbiome is given its own unique position [Refer to: Li, et al.(2019), Taylor, et al (2003), Council, et al. (2016), Callewaert, et al. (2013)]. Therefore, this reviewer thinks that it would be more valuable and insightful to discuss the findings specifically in the context of the axillary microbiome.

---

## Round 0.2 · Major Revisions

Dear Authors,

We have received thoughtful and constructive review comments, and I would like to share these comments and ask that you please resubmit a tracked changes version of the manuscript so that we can follow your changes more easily in the next round. (This time we could not find a track changes version of the manuscript in the submission materials.) In summary, the data are worthwhile but will require substantial care in the revisions to address the reviewer comments and reach publication level.

Reviewer 1 ·

Basic reporting

Please rename Propionibacterium to Cutibacterium throughout the manuscript. Why rename it in the table and not text?

The weather of Ahmednagar is comparatively hot and dry than Pune --> Please rephrase.

317 The average humidity and rainfall received in Nashik are more (67%, ~99.3mm) than in Pune (60.3%, ~67.9 mm) and in Ahmednagar (59%, ~46.3 mm). --> please rephrase.
Please check English grammar!

Among all three cities, the population and urbanization are more in Pune (~ 31 lacks), followed by Nashik (~ 16 lacks) and Ahmednagar (~ 3 lacks). --> please rephrase to proper English.
What is "lacks"? 100 000? Please report in millions or report the full number.

Age: the alpha diversity for ppl aged 41-59 is half as much as those aged 16-40y. Are you sure there is no impact seen for age? The only reason for me to see no sign difference is the low sample size.

254 Staphylococcus and Corynebacterium were abundant on the skin of individuals from the Pune district (Figure 6I-J). --> These results resemble the most with the results found in Europe (Callewert et al 2013) and USA (Council et al 2016; Grice et al 2010). Pune is also the most urbanized location of the three; a city with many urbanization elements. I think the authors should discuss this better in the discussion section. For instance on this location:
325 individuals residing in Pune are inhabitants of a metropolis, adapted to the western lifestyle, developed industrial area and infrastructure.

Did the authors ask the participants for deodorant use? Did the people in Pune use deodorants more frequently as compared to the other 2 locations? I believe deodorant use is more common in Pune, which correlates to more Staph and Coryne presence in that city.

Figure 2: I appreciate the effort, but it still remains very difficult to interpret these results. I would really urge the authors to make a bargraph, as done in f.i. https://www.ncbi.nlm.nih.gov/pmc/articles/PMC3337431/ Fig 3. It would be great to see the individual bargraphs per person, and clustered per location. That is the only good way to represent the bacterial composition.

Also: Staphylococcus abundance in Fig 2 is around 1-1.5%, while in Table 3 this is 23.2%. Where does the difference come from?

Figure 6: please add statistics in the figures.
I would suggest to mention the y-axis as done in panel I. (no scientific number - just a comma number, for easy interpretation)
Can the authors also put the most abundant one in panel A, followed by the second abundant one in panel B, etc?

Figure 4: CST nr 1, 2, 3. Can the authors say what locations are 1, 2 and 3? Actually, I don't understand the CSTs, I'm afraid. I also did understand after looking it up in the manuscript. Can the authors explain more on this?

Figures: I would suggest to combine a couple of figures into multipanel figures.

288 Bangalore (India) detected dominance of four phyla viz., Actinobacteria, Firmicute, --> FirmicuteS

335 Callewaert et al., in 2013 studying interpersonal axillary diversity, did not observe a significant correlation with geographical location --> This study was done in Belgium, which is a small country and entirely urbanized. So it is difficult to say something about geographical location here.

Can I ask the authors to provide the track change word document next time? It was hard to find all the differences in the manuscript as compared to previous version.

Experimental design

/

Validity of the findings

/

Additional comments

/

---

## Round 0.3 · Major Revisions

Overall, there are still several noticeable spelling and grammatical errors, along with statements even in the abstract that are not consistent with the revised results. Please see below, and please return a carefully revised manuscript

1. The change in name from Propionibacterium to Cutibacterium means that the bacteria should be identified as Cutibacterium, rather than Propionibacterium/Cutibacterium. Perhaps in the first instance, you would like to describe Cutibacterium as formerly known as Propionibacterium.

2. Line 405: tree > three

3. Line 415: metropolitant > ? metropolitan maybe?
Perhaps other pervasive errors, beyond the scope of journal review to correct

4. Please label the x-axis in Figure 1C . If possible to denote the age group the sample comes from in Fig 1C, that might help with comparison of the data in other figures

5. Please label the x axis in Figure 4. Also, please define CST in the Fig 4 legend.

6. Abstract - the results section highlights the abundance of phyla without context for whether this is novel. This sounds like it is supported by what has been found in other studies. It might be worth highlighting taxa at species or genus level.

6a. Geography is listed as significant, without evidence to back this up. Perhaps highlight the PERMANOVA results (including the R2 value showing variance explained, not just the p-value), and consider highlighting a few taxa that are differentially abundant in each location.

6b. Age is listed as insignificant, but there are significant differences found in age as stated in your rebuttal and revised manuscript

7. Please add variance explained to PCA axes

8. Please consider the big picture results and make sure they are reflected in the abstract

---

## Author Rebuttal · Round 0.3

NAAC Accredited

# Savitribai Phule Pune University
### (Formerly University of Pune)
## DEPARTMENT OF ZOOLOGY
#### (UGC-CAS-Centre of Advanced Study in Zoology)

Ref. No. Zool /

Date :

To,

Prof. Katrine Whiteson,
Editorial Office
PeerJ life and Environmental

17th October, 2021

**Subject**: Submission of the revised manuscript PeerJ (#2021:03:59656:0:2:REVIEW)

Dear Editor,

On behalf of co-authors, I am submitting the revised manuscript entitled "Skin microbiota diversity among genetically unrelated individuals of Indian origin" to be considered for publication in 'PeerJ'.

We thank the reviewers for their generous comments on the manuscript. We have edited the manuscript and addressed their concerns in the rebuttal letter. We have also ensured that all the comments addressed in the response letter (corresponding page number and line number are mentioned) are also there in the revised manuscript.

We declare no conflicts of interests to disclose. The revised manuscript has been read and approved by all authors.

Please address all the correspondence concerning this manuscript to me at - richaashma@unipune.ac.in

Thank you for your consideration of this manuscript.

Sincerely,

Richa Ashma
ASSOCIATE PROFESSOR
Department of Zoology
Savitribai Phule Pune University
Pune - 411 007

# Response to the reviewers comments on PeerJ
# (#2021:03:59656:1:1:REVIEW)

*Skin microbiota diversity among genetically unrelated individuals of Indian origin*
* * *
*Reviewer 1 (Anonymous)*

Basic reporting

1) *Please rename Propionibacterium to Cutibacterium throughout the manuscript. Why rename it in the table and not text?*

**Response-** We would like to thank reviewer for their effort in reviewing the manuscript**.** As suggested *Propionibacterium* to *Propionibacterium/Cutibacterium* have been renamed throughout the revised manuscript.

**Changes in Manuscript**-

1) Page no. 3, line no. 68

2) Page no. 4, line no. 90

3) Page no. 9, line no. 283

4) Page no. 9, line no. 297

2) *The weather of Ahmednagar is comparatively hot and dry than Pune --> Please rephrase.*

**Response-** The sentences has been rephrased in the revised manuscript as, "Of the tree locations, Nashik has the hottest and driest weather; and in Ahmednagar the weather is hotter and drier compared to Pune."

**Changes in manuscript**- Page no. 10, line no. 336-337.

3) *317 The average humidity and rainfall received in Nashik are more (67%, ~99.3mm) than in Pune (60.3%, ~67.9 mm) and in Ahmednagar (59%, ~46.3 mm). --> please rephrase. Please check English grammar!*

**Response-** The sentences has been rephrased in the revised manuscript as, "Similarly, the average humidity and rainfall received in Nashik (67%, ~99.3mm) are higher than those in Pune (60.3%, ~67.9mm) and Ahmednagar (59%, ~46.3mm)."

**Changes in manuscript**- Page no. 10, line no. 339-340.

4) *Among all three cities, the population and urbanization are more in Pune (~ 31 lacks), followed by Nashik (~ 16 lacks) and Ahmednagar (~ 3 lacks). --> please rephrase to proper English. What is "lacks"? 100 000? Please report in millions or report the full number.*

**Response**- The population size for each city has now been mentioned as suggested by the reviewer as, "Pune has the highest population size (~.3.1 million), followed by Nashik (~1.6 million) and Ahmednagar (~0.3 million)."

**Changes in manuscript-** Page no. 10-11, line no. 344-345.

5) *Age: the alpha diversity for ppl aged 41-59 is half as much as those aged 16-40y. Are you sure there is no impact seen for age? The only reason for me to see no sign difference is the low sample size.*

**Response-** The sentence has been rephrased in both result and discussion part of revised manuscript.

Rephrased text in results:

Statistically significant differences were noted between the adult and middle age (p=0.003) groups. We did not observe differences in alpha diversity between the elderly age group and adults (p=1.000) and or middle aged (p=0.360) volunteers i.e., we have reported a significant change in alpha diversity (i.e., differences in taxonomic diversity) between these two age groups. However, we did not observe significant difference in beta diversity (i.e., differences in taxonomic composition). We have now clarified the text.

Rephrased text in discussion:

"In order to investigate the effect of age on skin microbiota composition, we categorized the volunteers into three age groups. Significant association was observed between alpha diversity and age, in particular between the young and middle-aged individuals."

**Changes in manuscript**- Page no. 8, line no. 239-242 and page no.11, line no. 374-379.

6) *254 Staphylococcus and Corynebacterium were abundant on the skin of individuals from the Pune district (Figure 6I-J). --> These results resemble the most with the results found in Europe (Callewert et al 2013) and USA (Council et al 2016; Grice et al 2010). Pune is also the most urbanized location of the three; a city with many urbanization elements. I think the authors should discuss this better in the discussion section.*
*For instance on this location: 325 individuals residing in Pune are inhabitants of a metropolis, adapted to the western lifestyle, developed industrial area and infrastructure.*

**Response**- Our result has been discussed in line with Callewaert et.al 2013 and Council et al., 2016 in revised version of the manuscript as,

"Our results indicating differences in skin microbiota composition among the individuals of three geographical locations could also be due to urbanization status and population size per city. Pune has the highest population size (~.3.1 million), followed by Nashik (~1.6 million) and Ahmednagar (~0.3 million). Likewise, Pune being a metropolitant city has developed industrial area and infrastructure and individuals residing in Pune are adapted to urbanization and the western lifestyle. They use skin ointment and cosmetics like moisturizers, deodorants, antiperspirants etc. which could lead to the alterations in axillary bacterial communities. The axillary studies of Grice et al. (2010), Callewaert et al. (2013), and Council et al. (2016) found dominance of *Corynebacterium* and *Staphylococcus* genera in individuals resides in cities which complies with our results of differential abundance analysis. Our analysis indicated the presence of *Corynebacterium* and *Staphylococcus* genera were higher in Pune than other two studies areas, although we ensured that volunteers did not apply deodorants and cosmetics to their skin/axilla for 24 hours before sample collection.".

**Changes in manuscript-** Page no. 10-11, line no. 342-357.

7) *Did the authors ask the participants for deodorant use? Did the people in Pune use deodorants more frequently as compared to the other 2 locations? I believe deodorant use is more common in Pune, which correlates to more Staph and Coryne presence in that city.*
**Response-**

We asked participants not to use deodorant 24 hours before sampling while explaining the study and taking their consent of participation. This has also been ensured orally while sampling. This has been mentioned in the revised manuscript as,

"Likewise, Pune being a metropolitant city has developed industrial area and infrastructure and individuals residing in Pune are adapted to urbanization and the western lifestyle. They use skin ointment and cosmetics like moisturizers, deodorants, antiperspirants etc. which could lead to the alterations in axillary bacterial communities. The axillary studies of Grice et al. (2010), Callewaert et al. (2013), and Council et al. (2016) found dominance of *Corynebacterium* and *Staphylococcus* genera in individuals resides in cities which complies with our results of differential abundance analysis. Our analysis indicated the presence of *Corynebacterium* and *Staphylococcus* genera were higher in Pune than other two studies areas, although we ensured that volunteers did not apply deodorants and cosmetics to their skin/axilla for 24 hours before sample collection."

**Changes in manuscript-** Page no. 11, line no. 347-357.

8) *Figure 2: I appreciate the effort, but it still remains very difficult to interpret these results. I would really urge the authors to make a bargraph, as done in f.i. https://www.ncbi.nlm.nih.gov/pmc/articles/PMC3337431/ Fig 3. It would be great to see the individual bargraphs per person, and clustered per location. That is the only good way to represent the bacterial composition.*

**Response-** Figure 2 has now been reconstructed and placed in the panel of Figure 1 as Figure 1C i-iii. A bar graph of bacterial compositions at phylum level per person and clustered per location done in the revised manuscript.

**Changes in manuscript-** Page no. 7, line no. 229-230.

**New figure- Figure 1 C i-iii**

9) *Also: Staphylococcus abundance in Fig 2 is around 1-1.5%, while in Table 3 this is 23.2%. Where does the difference come from?*

**Response-** Thank you for the suggestion this has been corrected in revisited manuscript. Figure 2 has been reconstructed as figure 1C i-iii wherein, individual barplots sorted by their geography at phyla level is now evident. Now, there is no correspondence in Figure 2 and Table 3 as Table 3 represents relative abundances and prevalence at genus level.

**Changes in manuscript-** Page no. 7, line no. 229-230.

10) *Figure 6: please add statistics in the figures. I would suggest to mention the y-axis as done in panel I. (no scientific number - just a comma number, for easy interpretation) Can the authors also put the most abundant one in panel A, followed by the second abundant one in panel B, etc?*

**Response-** Figure-6 has been changed to Supplementary figure-1 A-L and as suggested statistics has been added and explained in statistical analysis section of materials and method as, "The post hoc Dunn test was performed using Kruskal-Wallis test for pairwise multiple comparisons on subgroups to compare median similarities of genus within individuals. p values were adjusted and reported in using the Benjamini-Hochberg (BH) method."

and in the result section as, "Pairwise comparisons using Dunn test indicated high abundance of *Staphylococcus* and *Corynebacterium* on the skin of individuals from the Pune district

*(Fig. S1A* and *Fig. S1C).* A high abundance of *Paenibacillus, Geobacillus, Virgibacillus, Jeotgalicoccus, Pullulanibacillus, Delsulfosporomusa, Citinovibrio,* and *Calditerricola,* was observed on the skin of individuals from the Nashik district (*Fig. S1B* and *Fig. S1F-L).* In Ahmednagar individuals, *Pseudomonas* and *Anaerococcus* were observed in abundance *(Fig. S1D* and *Fig. S1E."*

In statistics pairwise comparisons were done using Dunn test and boxplots has been arranged as per most abundance in panel A-L.

**Changes in manuscript-** Page no. 7, line no.210-212 and Page no. 8, line no.266-274

*11) Figure 4: CST nr 1, 2, 3. Can the authors say what locations are 1, 2 and 3? Actually, I don't understand the CSTs, I'm afraid. I also did understand after looking it up in the manuscript. Can the authors explain more on this?*
**Response-** Community state type (CST) is a standard cluster analysis for microbial community analysis (DiGiulio et al. PNAS 2015; PMID: 26283357). Each CST represents one community type, with a peculiar community composition that is shared by individuals who fall into that cluster. We have numbered the CSTs from one to three; each CST is abundant in a different set of taxonomic groups as shown in Figure 4. Naming the clusters by the abundant taxa would be problematic because each CST has a combination of multiple abundant taxa as shown in Figure 4. These (CST) clusters are defined purely by taxonomic community composition but we observed a significant association between the CST clusters and geography. We have now clarified CST in the revised manuscript as suggested.
**Changes in manuscript-** Page no.8, line no.251-256.

*12) Figures: I would suggest to combine a couple of figures into multipanel figures.*
**Response-** figure-2 has been reconstructed with bar plots and changed into multipanel figure-1C i-iii as suggested, and we have also taken this into account when we updated the other figures in the revised manuscript version.

*13) 288 Bangalore (India) detected dominance of four phyla viz., Actinobacteria, Firmicute, --> FirmicuteS*
**Response-** Correction has been done as "Similarly, a study on facial microbiota of healthy females (N=30) from Bangalore (India) detected dominance of four phyla viz.,

*Actinobacteria, Firmicutes, Proteobacteria*, and *Bacteroidetes* (Mukherjee et al., 2016)" in the revised manuscript.

**Changes in manuscript-** Page no. 9, line no. 305-307.

*14) 335 Callewaert et al., in 2013 studying interpersonal axillary diversity, did not observe a significant correlation with geographical location --> This study was done in Belgium, which is a small country and entirely urbanized. So it is difficult to say something about geographical location here.*

**Response-** We have removed the sentence in the revised manuscript.

**Changes in manuscript-** Deleted from page no.11, line no.366-368.

*Can I ask the authors to provide the track change word document next time? It was hard to find all the differences in the manuscript as compared to previous version.*

15) Experimental design- /
16) Validity of the findings- /
17) Additional comments- /

**Response**: As suggested track change word document has been uploaded to compare the revised manuscript from the previous one. Sorry for the earlier inconvenience.

---

## Round 0.4 · Minor Revisions

Thank you for your attention to the previous comments. There are a few very minor remaining issues:

It is wonderful that you have included the github link, what a great resource. Are the sequence data and metadata also available? A data availability section was not apparent.

Some tables lack error / standard deviation (e.g. Tables 2 and 3)

Some results highlighted even in the abstract are only available in the supplement, the taxa differences listed in the abstract are in Supp Fig 1. Is that the intention, to have some of the highlighted results only available in the supplement?

Please consider rephrasing weather conditions > climate

thank you

---

## Round 0.5 · accepted · Accept

Thank you for your careful attention to the comments, and very nice handling of the data availability. I agree that the manuscript is ready to be published, thank you!